# Intravitreal Administration of Retinal Organoids-Derived Exosomes Alleviates Photoreceptor Degeneration in Royal College of Surgeons Rats by Targeting the Mitogen-Activated Protein Kinase Pathway

**DOI:** 10.3390/ijms241512068

**Published:** 2023-07-27

**Authors:** Jung Woo Han, Hun Soo Chang, Jin Young Yang, Han Sol Choi, Hyo Song Park, Hyoung Oh Jun, Ji Hye Choi, Sun-Sook Paik, Kyung Hwun Chung, Hee Jeong Shin, Seungyeon Nam, Ji-Hye Son, Si Hyung Lee, Eun Jung Lee, Kyoung Yul Seo, Jungmook Lyu, Jin Woo Kim, In-Beom Kim, Tae Kwann Park

**Affiliations:** 1Department of Ophthalmology, Soonchunhyang University Bucheon Hospital, Bucheon 31538, Republic of Korea; 106236@schmc.ac.kr (J.W.H.); 121966@schmc.ac.kr (H.S.C.); 124533@schmc.ac.kr (H.S.P.); sieh12@schmc.ac.kr (S.H.L.); 2Department of Microbiolo and BK21 FOUR Project, Soonchunhyang University College of Medicine, Cheonan 31538, Republic of Korea; hschang@sch.ac.kr (H.S.C.); son310321@naver.com (J.-H.S.); 3Laboratory of Molecular Therapy for Retinal Degeneration, Soonchunhyang University Bucheon Hospital, Bucheon 31538, Republic of Korea; roswellgirl111@gmail.com (J.Y.Y.); jhonor@daum.net (H.O.J.); wisdomhhh@gmail.com (J.H.C.); cyung6767@gmail.com (K.H.C.); 4Department of Anatomy, College of Medicine, The Catholic University of Korea, Seoul 14662, Republic of Korea; paikss@catholic.ac.kr (S.-S.P.); ibkimmd@catholic.ac.kr (I.-B.K.); 5Catholic Institute for Applied Anatomy, College of Medicine, The Catholic University of Korea, Seoul 14662, Republic of Korea; 6Department of Interdisciplinary Program in Biomedical Science, Soonchunhyang Graduate School, Soonchunhyang University Bucheon Hospital, Bucheon 31538, Republic of Korea; shj3277@naver.com; 7Department of Neuroscience and Behavior, University of Notre Dame College of Science, Notre Dame, IN 46556, USA; snam3@nd.edu; 8Department of Biological Sciences and KAIST Stem Cell Center, Korea Advanced Institute of Science and Technology, Daejeon 34141, Republic of Korea; ieunjung@kaist.ac.kr (E.J.L.); jinwookim@kaist.edu (J.W.K.); 9Department of Ophthalmology, Severance Hospital, Institute of Vision Research, Yonsei University College of Medicine, Seoul 03722, Republic of Korea; seoky@yuhs.ac; 10Department of Medical Science, Konyang University, Daejun 32992, Republic of Korea; lyujm@konyang.ac.kr; 11oligoNgene Pharmaceutical Co., Ltd., Bucheon 31538, Republic of Korea

**Keywords:** retinal organoids, extracellular vesicles, exosomes, retinal degeneration, MAPK pathway signaling

## Abstract

Increasing evidence suggests that exosomes are involved in retinal cell degeneration, including their insufficient release; hence, they have become important indicators of retinopathies. The exosomal microRNA (miRNA), in particular, play important roles in regulating ocular and retinal cell functions, including photoreceptor maturation, maintenance, and visual function. Here, we generated retinal organoids (ROs) from human induced pluripotent stem cells that differentiated in a conditioned medium for 60 days, after which exosomes were extracted from ROs (Exo-ROs). Subsequently, we intravitreally injected the Exo-RO solution into the eyes of the Royal College of Surgeons (RCS) rats. Intravitreal Exo-RO administration reduced photoreceptor apoptosis, prevented outer nuclear layer thinning, and preserved visual function in RCS rats. RNA sequencing and miRNA profiling showed that exosomal miRNAs are mainly involved in the mitogen-activated protein kinase (MAPK) signaling pathway. In addition, the expression of MAPK-related genes and proteins was significantly decreased in the Exo-RO-treated group. These results suggest that Exo-ROs may be a potentially novel strategy for delaying retinal degeneration by targeting the MAPK signaling pathway.

## 1. Introduction

Retinal degeneration (RD), including retinitis pigmentosa, remains the primary cause of irreversible blindness worldwide [1,2,3]. Photoreceptor cell death from RD causes significant visual loss in retinal degenerative diseases [4,5,6]. Although many pathways have been implicated and the precise mechanisms of photoreceptor death are not well understood, some known processes include oxidative stress, inflammation, endoplasmic reticulum stress, and direct rod mutation-dependent effects [7,8,9]. However, despite advances in our knowledge of the pathogenesis of RD, the precise mechanism underlying subsequent photoreceptor cell death remains unknown. Therefore, therapeutic strategies targeting the various mechanisms that trigger photoreceptor cell death pathways may have the benefit of broader clinical applications than for treating RD. Recently, much attention has been paid to identifying these therapeutic approaches for RD.

Exosomes are extracellular vesicles (EVs) which are approximately 100 nm in size, with a lipid membrane and cargo of microRNAs (miRNAs), messenger RNAs (mRNAs), lipids, and various proteins. They are secreted by all cell types and are involved in intercellular communication via protein and RNA delivery [10,11,12]. Exosomes are being increasingly studied as important indicators of retinopathies, including their involvement in retinal cell degeneration upon insufficient release [13]. The exosomal transfer of miRNAs has been extensively reported to mediate the effects of EVs in treating a range of disorders. Individual miRNAs have been found to play crucial roles in regulating ocular and retinal cell functions, including those essential for photoreceptor maturation, maintenance, and visual function [14,15,16,17]. In addition, miRNA dysregulation is associated with photoreceptor death in the degenerating retina [18,19]. A recent study reported that intravitreal mesenchymal cell transplantation prevented photoreceptor death and protected retinal function by mediating miRNA-21 in a mouse model of photoreceptor loss induced by N-methyl-N-nitrosourea [20]. However, few studies have attempted to elucidate the role of miRNAs in regulating photoreceptor cell death.

Organoids are used to create in vitro tissues that mimic their natural counterparts [21,22,23,24,25]. Several studies have developed retinal organoids (ROs) that closely resemble many aspects of the real retina, using induced pluripotent stem cells (iPSCs) or embryonic stem cells [26,27]. ROs can be used as human retinal implants or in gene therapy, drug screening, and studying RD [28]. Zhou et al. [29] recently demonstrated that human iPSC (hiPSC)-derived ROs release exosomes and microvesicles which contain small non-coding RNAs. Although studies have investigated exosomes derived from many different cell types [30], there has been no study on the therapeutic effects of RO-derived exosomes on RD.

The mitogen-activated protein kinase (MAPK) signaling pathway, an important cell signaling pathway, plays an important regulatory role in pathological processes, including cell proliferation, necrosis, inflammation, and apoptosis [31,32]. Cell death is known to trigger multiple MAPK pathways [33]. For example, previous studies have reported that p44/42 is a major pro-apoptotic transcriptional factor for retinal ganglion cells [34] and inhibiting the p44/42 pathway protects against damage resulting from ischemic injury of the brain [35]. In addition, inhibition of phospho-p38 (p-p38) reduced light-induced photoreceptor apoptosis [36]. Therefore, inhibiting the MAPK pathway may be meaningful because it has a therapeutic effect on reducing cell death.

In the present study, we established ROs using hiPSCs and extracted exosomes from RO (Exo-ROs) differentiation medium conditioned for 60 days (D60). We demonstrated that intravitreal administration of Exo-ROs markedly alleviated photoreceptor apoptosis and preserved retinal function in the Royal College of Surgeons (RCS) rats. Furthermore, the expression of MAPK-related genes and proteins was significantly decreased in Exo-RO-treated retinas. In addition, exosomal miRNAs from Exo-ROs are mainly involved in the MAPK signaling pathway. These results suggest that targeting the MAPK signaling pathway and focusing on Exo-ROs may be a potentially novel strategy for delaying RD.

## 2. Results

### 2.1. Generation of hiPSC-Derived ROs and Characterization of Exosomes in the hiPSC-Derived ROs-Conditioned Medium

Based on a previously published protocol [37], we successfully generated ROs from hiPSCs. Figure 1A shows a schematic of RO differentiation. The hiPSC-derived aggregates were obtained gradually form EBs. The optic vesicles were collected manually on days 25 to 28 and formed 3D ROs when cultured in suspension. After 28 days of differentiation, the ROs showed unique morphological and phenotypic characteristics, forming a bright, stratified layer toward the periphery (Figure 1B). At 60 days of differentiation, ROs exhibited widespread expression of the retinal progenitor marker CHX10, the proliferation marker Ki67 within the outer layer, and the ganglion cell marker HuC/D within the inner layers of ROs (Figure 1C). The conditioned medium was collected for exosome isolation.

To determine whether hiPSC-derived 3D ROs release exosomes, we analyzed the RO-conditioned medium at D60. The EVs were successfully isolated and analyzed using NTA to determine their size and concentration (Figure 1D). The average EV diameter was 149.5 ± 2.9 nm, with an average peak diameter of 72 ± 1.8 nm. While the EVs from ROs contained both exosomes and microvesicles, most EVs corresponded to exosomes with diameters of 30 to 150 nm. The concentration of released EVs was 4.54 × 10^10^ ± 5.86 × 10^8^ particles/mL. The Exo-ROs were further characterized using transmission electron microscopy, which revealed that they were cup-shaped, with an average diameter of approximately 110 nm (Figure 1E). Immunoblot analysis showed that the levels of exosome markers CD9, CD63, and CD81 were higher in the isolated exosomes than in the crude precipitate of the RO-conditioned medium. In contrast, the cell-specific markers β-actin and calnexin were not detected in the isolated exosomes (Figure 1F).

### 2.2. Exo-ROs Alleviate Photoreceptor Cell Apoptosis and Preserve Visual Function in RCS Rats

To determine how the Exo-ROs affected the photoreceptors in RCS rats, we intravitreally injected 2 uL of concentrated Exo-RO solution (containing 1.0 × 10^10^ particles/mL) into each RCS rat’s eye. In group 1, Exo-ROs were injected on a PND 21, whereas in group 2, they were injected twice at PNDs 21 and 35 to investigate the therapeutic effect of repeated injections in the late stages of RD. We then investigated the effects of Exo-ROs 14 days after injection (Figure 2A).

Additionally, we investigated the effects of Exo-ROs on photoreceptor apoptosis using TUNEL staining. Many of the TUNEL-positive cells were detected in the ONL of the untreated retina and vesicle-treated eyes, whereas the number of TUNEL-positive cells in the treated eyes was significantly decreased in the Exo-ROs treated retinas of group 1 and 2 (Figure 2B,C,E,F). The ONL thickness was significantly preserved in Exo-RO-treated eyes in group 1 and 2 compared with that in the untreated and vesicle-treated eyes (Figure 2D,G) as well. In detail, ONL thickness was preserved more in the central region of the retinas than in the peripheral region of the retinas in group 1. However, in group 2, ONL thickness was preserved similarly not only in the central retinas but also in the peripheral retinas.

To investigate whether Exo-ROs improve visual function, the scotopic ERG and OMR tests were performed using groups 1 and 2 for 5 and 7 weeks, respectively. In accordance with the morphological modifications of the photoreceptor cells, Exo-ROs treatment led to the preservation of both scotopic a- and b-waves when compared with the control and vesicle-treated groups (Figure 3A). The RCS rats were also examined with the OMR test, which measured visual acuity by assessing the motor response under photopic conditions and with virtual rotation of stripes of different widths. OMR also showed that Exo-RO treatment significantly preserved the visual acuity of RCS rats compared to the vesicle-treated or untreated controls, which showed similar visual acuity (Figure 3B). These results demonstrated that Exo-ROs alleviated photoreceptor apoptosis and rescued visual function in RCS rats.

To evaluate the morphological effects of Exo-ROs on photoreceptor cells in degenerating retinas, transverse retinal sections were immunostained with the photoreceptor markers recoverin; rhodopsin; and M/L- and S-opsins. Strong recoverin and rhodopsin immunoreactivities were observed throughout the ONL, including the inner and outer segments, within the Exo-RO-treated groups relative to the control or vesicle-treated ones. The fluorescence intensities of rhodopsin and recoverin were also significantly higher in Exo-RO-treated retinas than in the control or vesicle-treated retinas of both groups (Figure 4A,B,D,E). In addition, the analysis of the amounts of M/L- and S-opsins in the control, vesicle-, and Exo-RO-treated retinas showed that the numbers of M/L- and S-opsins were significantly increased in the Exo-RO-treated retinas of both groups (Figure 4A,C,D,F).

### 2.3. Changes in the Gene Expression Profile of RCS Rats after Exo-ROs Treatment

To determine the mechanism underlying photoreceptor cell apoptosis, we profiled the total RNAs in the eyes of vesicle-treated and Exo-RO treated RCS rats using RNA-seq. In the total RNA-seq of the eyes of RCS rats, Exo-RO treatment significantly upregulated the expression of retinal cell markers, including the photoreceptors and RPE cells, compared with that in the vesicle treatment (Figure 5A). There were 189 genes upregulated in the Exo-RO-treated eyes relative to those in vesicle-treated eyes (fold change > 1, false discovery rate (FDR) *q* < 0.05). In the gene ontology analysis, the genes were enriched in ‘endocrine system development’, ‘response to retinoic acid’, and ‘visual perception’ for biological processes (Figure 5B). The ‘extracellular matrix’ (38.46%) and ‘photoreceptor outer segment’ (17.31%) for cellular components (Figure 5C) indicated that Exo-RO treatment rescued photoreceptors and improved retinal function. Our data showed that the downregulated genes were involved in the pathways regulating cell proliferation, differentiation, and survival, including MAPK, PI3K-Akt, and RAS signaling in Exo-RO-treated retinas compared with that in vesicle-treated retinas (Figure 5D).

### 2.4. Analysis of miRNA Expression in Exo-ROs

To determine the miRNA content of Exo-ROs, RNA-seq profiling was performed. In Exo-ROs, 128 miRNAs with more than 100 counts per million (cpm) were detected. Among them, the 20 most abundant miRNAs with more than 1000 cpm are shown in Figure 6A. In target gene prediction, 4929 human genes were targeted by more than 3 of the 20 most abundant miRNAs. The biological processes of 293 genes that were targeted by 10 or more miRNAs were enriched in functions such as the ontology terms ‘regulation of cellular amide metabolic process’, ‘regulation of cell development’, and ‘response to transforming growth factor beta’ (Figure 6B). In pathway enrichment analysis, the genes were enriched in ‘pathways in cancer’, ‘MAPK signaling pathway’, and ‘signaling pathways regulating pluripotency of stem cells’, (Figure 6C) of which the key pathway was that for MAPK signaling (Appendix A). To evaluate the roles of the target genes regulated by Exo-RO miRNAs, we compared the predicted target genes and DEGs in exosome-treated RCS rats. Among the 4664 rat orthologs of the 4929 human predicted target genes, 301 genes were significantly downregulated in Exo-RO-treated eyes compared with those in vesicle-treated eyes (*p* < 0.05; Figure 6D). The genes were mainly enriched in the MAPK signaling pathway (61.22%) (Figure 6E and Appendix A). The genes enriched in the MAPK pathway were DUSP16, EGFR, FOS, MAP2K4, MAPK1, MAPK8, MAPT, NF1, RASA2, RASGRP1, RPS6KA5, and TEK in Exo-ROs (Figure 6F). Of the 20 most abundant miRNAs in Exo-RO, 15 miRNAs were predicted to target these 12 genes (Appendix A). Among these, hsa-let-7b-5p could target 11 of the 12 genes, suggesting that hsa-let-7b-5p may be a key miRNA for regulating MAPK pathway.

### 2.5. Exosomal miRNA from hiPSC-Derived ROs Inhibits MAPK Signaling in RCS Rats

To confirm that Exo-ROs led to significant MAPK pathway downregulation, reverse transcription quantitative PCR was performed. The expression of MAPK-related genes included *MAP2K4*, *MAPK1*, *MAPK8*, *MAPK10*, *c-FOS*, and *c-JUN*, and these were significantly lower in Exo-RO-treated retinas than that in the control and vesicle-treated retinas (Figure 7A). The levels of the MAPK pathway markers (p-p38, p38, p-JNK, JNK, p-ERK, and ERK) were determined using Western blotting. The expression of p-JNK/JNK and p-ERK/ERK were significantly decreased in Exo-RO-treated retinas relative to those in the control and vesicle-treated retinas (Figure 7B,C).

MAPK activation, represented by the expression of p-p44/42, p-JNK, and p-p38, was determined using immunofluorescence. In control and vesicle-treated retinas, the expressions of p-p44/42, p-JNK, and p-p38 were predominantly expressed in the photoreceptor’s inner and outer segments but weakly in the Müller cells, inner nuclear layer, and retinal ganglion cells. We found that the expressions of all three genes were significantly decreased in Exo-RO-treated retinas in both groups (Figure 8A–D).

### 2.6. Confirmation of the Effect of Exo-ROs in an Oxidative Stress Model of RPE Cells

To confirm the effect of Exo-ROs using another in vitro model, and closely following the vivo experiments, primary rat RPE cells were treated with a specific H_2_O_2_ concentration to establish an RPE cell model of oxidative stress injury. To examine the uptake of exosomes, calcein-labeled Exo-ROs were incubated with cultured RPE cells for 30 min. Green fluorescein particles were observed throughout the cell cytoplasm (Figure 9A). Exo-ROs treatment significantly suppressed the expression of p-JNK/JNK and p-ERK/ERK compared to that of the H_2_O_2_-treated group (Figure 9B,C). Furthermore, we investigated the anti-apoptotic effect of Exo-ROs. As expected, the expression level of cleaved caspase-3 was significantly inhibited by the treatment with Exo-ROs compared to that of the H_2_O_2_-treated group (Figure 9B,C).

## 3. Discussion and Conclusions

Photoreceptor death has been suggested as the final common pathway of retinal degenerative diseases and a cause of visual loss with few treatments available. In the present study, we used scotopic ERG and OMR tests to demonstrate that intravitreal Exo-RO injections led to retinal function rescue in RCS rats. Morphologically, Exo-ROs alleviated photoreceptor cell death and delayed ONL thinning. RNA sequencing and miRNA profiling showed that exosomal miRNAs released from Exo-ROs mainly targeted the MAPK signaling pathway. Subsequently, the therapeutic effect of Exo-ROs was further reconfirmed using the H_2_O_2_-mediated oxidative stress model of RPE cells, which showed that Exo-ROs alleviated cell death via inhibiting the MAPK signaling pathway. Our results provided the first evidence that exosomes derived from ROs can treat RD.

Recent statistical studies and advancements in cell-based therapies complement traditional approaches in the treatment of ocular diseases. These therapies hold immense potential for restoring vision and enhancing the quality of life for patients. The transplantation of stem cells is one of the potential therapeutic options for rescuing vision loss associated with RD [38]. However, considering the safety concerns, including cellular proliferation, malignant transformation, and immune rejection, using exosomes isolated from a conditioned medium is a potentially safe alternative to cell therapy. Furthermore, exosomes can easily penetrate physical barriers and enter target tissues [39]. A previous study reported that the direct administration of exosomes from neural progenitor cells in the subretinal space of RCS rats delayed photoreceptor degeneration and preserved visual function for 28 days after injection [40]. A subretinal injection is widely used in scientific research and clinical applications and is regarded as one of the best strategies for gene therapy or stem cell transplantation [41,42]. However, it is technically more demanding than intravitreal injection and is difficult to perform precisely and repeatedly. In addition, the effect of the drug appears only in the localized area around the injection site [43]. Here, we showed that intravitreal injection of Exo-ROs sufficiently inhibited photoreceptor cell death throughout the entire retina in RCS rats. In addition, we performed a preliminary study to determine the interval of Exo-RO injection. As a result, we established that a significant rescue of the photoreceptors was observed after 14 days of injections. This result led us to inject exosomes intravitreally every two weeks to sustain their therapeutic effects. The repeated injection group (group 2) exhibited significantly reduced photoreceptor cell death and preserved visual function compared with the vesicle-treated or untreated control group. However, there is a limitation to consider when interpreting the results of this study. Despite the few studies on the therapeutic effect of exosomes [44,45,46], positive control was not used to compare the effects of Exo-ROs due to differences in the origin of exosomes and administration route of injection. Further in-depth studies are required to address this issue.

Several studies have focused on the therapeutic effects of exosomes in vitro and in vivo models of retinal diseases [47,48]. The effects of exosomes on their target tissues are variable because the contents of exosomes are secreted from different cell types or under different conditions, as these may vary [49,50]. In particular, mesenchymal stem cells (MSCs) have been extensively studied in various ocular diseases, and MSC-derived exosomes promote retinal ganglion cell survival and preserve their functions [51,52,53,54,55]. In another study, MSC-derived exosomes effectively reduced the expression of inflammatory cytokines and ameliorated diabetic retinopathy progression in the retinas of diabetic rats [44,56]. Moreover, MSCs play an important role in wound modulation in retinal injuries [45,46]. Nevertheless, MSCs have distinctive mesenchymal lineages that are derived from the mesoderm. Given that the retina is derived from optic vesicles that arise from the neuroectoderm of the diencephalon, MSC-derived exosomes may have limited efficacy in the retina. In addition, RO differentiation recapitulates each of the main steps of human retinal development and contains all major retinal cell types. Exosomes from ROs are involved in the mechanisms of retinogenesis relevant to the developmental time points analyzed, and these correspond to the hallmarks of human retinal cell differentiation in vivo, indicating that they secrete various types of exosomes related to retinal progenitor cell proliferation, differentiation, and maintenance. Therefore, we postulate that exosomes derived during RO differentiation have a greater therapeutic effect on retinal cells via different mechanisms, representing a better option for RD therapy.

Our data showed that the downregulated genes were involved in pathways regulating cell proliferation-, differentiation-, and survival-related pathways, including MAPK, PI3K-Akt, and RAS signaling. MAPK signaling is significantly regulated by Exo-ROs. Therefore, we focused on the differential expression of MAPK signaling pathway genes in Exo-RO- and vesicle-treated retinas. Members of the MAPK family play important roles in many cellular processes, including cell proliferation, differentiation, survival, and apoptosis [31]. They are activated by diverse extracellular stimuli, including oxidative stress, heat shock, ultraviolet irradiation, and hypoxia [57,58,59]. Therefore, several studies have attempted to use MAPK inhibitors to treat retinal disease; for example, JNK inhibition to decrease apoptosis, vascular endothelial growth factor (VEGF) expression, and *choroidal neovascularization (CNV)* reduction in a laser-induced age-related macular degeneration model [60]. Moreover, regorafenib, a multi-kinase inhibitor targeting VEGF and Raf/MEK/ERK signaling, has been studied as a potential topical therapy in a nonhuman primate laser induced CNV model [61]. Unfortunately, MAPK inhibitors, which are currently used clinically, have difficulty treating retinal diseases because of their ocular toxicity [62,63,64]. Nonetheless, we demonstrated that Exo-RO treatment inhibited activated MAPK signaling in vitro and in vivo, suggesting that Exo-RO is an alternative treatment to MAPK inhibitors.

Exosomal miRNAs derived from MSCs play various roles in retinal development and function, including proliferation, differentiation, and death of radial peripapillary capillaries; differentiation of RPE cells; photoreceptor apoptosis; and microglial activation [65]. Although some miRNAs have been reported to exert their function via the MAPK/Erk2 pathway (for miR-133) [66] or the HMGB1 signaling pathway (for miR-126) [44], the mechanisms underlying the modulation of retinal functions are largely unknown. Of the top 20 abundant miRNAs in Exo-ROs, 16 have been known to be expressed in human or mouse retina and have their functions in retinal biology [65]; however, the remaining four miRNAs (miR-122-5p, miR-3591-3p, miR-128-3p, and miR-4443) were uniquely identified in our study, suggesting that they may be RO-specific exosomal miRNAs. Among them, miR-122-5p reportedly regulates the proliferation and apoptosis of chicken granulosa cells of hierarchal follicles by targeting MAPK3 [67] while the miR-128-3p-loaded nanocomplex enhances the chemotherapeutic effect of colorectal cancer through dual-targeting silencing of the PI3K/AKT and MEK/ERK pathways [68]. Since their effects on retinal biology are unknown, the roles of these RO-specific miRNAs in retinal development and function should be evaluated in further studies.

In conclusion, this study analyzed the therapeutic effects of RO-derived exosomes in RCS rats in an inherited RD model. We demonstrated that Exo-ROs alleviated photoreceptor cell apoptosis, altered retinal thickness (ONL), and preserved visual function by inhibiting the MAPK pathway. In addition, intravitreal exosome injections might be a non-cellular approach without the potential safety concerns associated with cell therapy, such as immune rejection. Collectively, our findings suggest that Exo-ROs may provide a platform for developing safe and effective therapeutics for RD treatment.

## 4. Materials and Methods

### 4.1. Animals

All animal experiments were conducted in accordance with the Guide for the Care and Use of Laboratory Animals and the Statement for the Use of Animals in Ophthalmic and Vision Research from the Association for Research in Vision and Ophthalmology. Moreover, all experiments were approved by the Institutional Animal Care and Use Committee of Soonchunhyang University Hospital, Bucheon, Korea (Project identification # SCHBCA2022-08; approval date 11 November 2022). The RCS/kyo rats (*rdy/rdy, p/p*) were purchased from Tadao Serikawa Graduate School of Medicine, Kyoto University. All RCS/kyo rats had free access to food and water and were raised under standardized conditions in a specific pathogen-free room with 12/12-h light/dark cycles. Anesthesia was induced via intraperitoneal injection of a mixture of Zoletil 50 (Virbac, Carros Cedex, France) and Rompun (Bayer Healthcare, Leverkusen, Germany) at a weight ratio of 4:1, and pupil dilation was performed with a mixture of 0.5% (*w*/*v*) tropicamide and 0.5% (*w*/*v*) *phenylephrine* (Hanmi Pharm, Seoul, Republic of Korea) prior to all examinations.

### 4.2. hiPSC Cultures

The American Type Culture Collection (ATCC) DYR0100 hiPSCs (ACS-1011; ATCC, Manassas, VA, USA) were used in our experiments. The hiPSCs were maintained in Essential 8 (E8) medium (Gibco^TM^; Thermo Fisher Scientific, Waltham, MA, USA) on culture dishes coated with vitronectin (Thermo Fisher Scientific). The cells were routinely cultured at 37 °C in a standard 5% CO_2_/95% air incubator with daily medium changes. Upon reaching approximately 70% to 80% confluency, the cells were mechanically passaged with the enzyme-free reagent ReLeSR (StemCell Technologies, Vancouver, BC, Canada) every 5 to 7 days. After that, the detached cell aggregates were collected in an E8 medium and carefully pipetted up and down to obtain a uniform suspension of cell aggregates, which were related at 1/10 to 1/60, depending on the confluence.

### 4.3. Differentiation into Three-Dimensional ROs and Conditioned Media Sample Collection

The ROs were differentiated from ATCC DYR0100 hiPSCs according to a retinal differentiation protocol recently described by Lee et al. [37]. The hiPSCs were maintained on vitronectin-coated culture dishes with E8 medium (Thermo Fisher Scientific) and dissociated by treatment with ReLeSR (StemCell Technologies). Next, the dissociated cells were plated on a low-attachment 6-well plate containing E8 medium (Thermo Fisher Scientific) with 3 µM ROCK inhibitor Y27632 (Tocris Biosciences, Abingdon, UK) and 3 µM Blebbistatin (Tocris Biosciences) at day 0 to induce embryoid body (EB) formation. Subsequently, the EBs were gradually replaced with neural induction medium (NIM) containing Dulbecco’s Modified Eagle Medium/Nutrient Mixture F-12 (DMEM/F12; 1:1; Thermo Fisher Scientific), 1% N-2 supplement (Thermo Fisher Scientific), non-essential amino acids (NEAAs), and 2 µg/mL heparin (StemCell Technologies) from E8 medium without ROCK inhibitor Y27632 and Blebbistatin (Tocris Biosciences). The day of detachment was annotated as day 0, with the medium being changed on days 1 (25% NIM), 2 (50% NIM), and 3 (100% NIM). On day 7, the EBs were plated on 35-mm Matrigel-coated dishes (Corning Life Sciences, Tewksbury, MA, USA) containing NIM at a density of 150 EB per dish. On day 15, the medium was switched from NIM to retinal differentiation medium (RDM), consisting of DMEM/F12 (3:1), 2% B-27 supplement without vitamin A (Thermo Fisher Scientific), NEAAs, and antibiotic-antimycotic solution (Thermo Fisher Scientific), and was changed every other day. On days 25 to 28, loosely adherent central portions of the neural clusters were lifted using a P1000 pipette under an Evos XL cell imaging microscope (*Invitrogen*, Waltham, MA, USA). The selected optic vesicle-like aggregates were further cultured to form three-dimensional (3D) ROs with RDM, supplemented with 10% exosome-depleted fetal bovine serum (FBS; Cat. No. A2720801; Thermo Fisher Scientific), 100 mM taurine (Sigma-Aldrich, St. Louis, MO, USA), and 2 mM L-alanyl-L-glutamine dipeptide (GlutaMAX^TM^; Thermo Fisher Scientific). For long-term organoid culture, the medium was changed every 3 days until the desired stage was reached. On day 60, the medium in which five organoids were cultured per milliliter was collected to obtain Exo-ROs. Thereafter, the collected media were centrifuged at 3000× *g* for 10 min to remove cell debris and frozen at −80 °C until further analysis.

### 4.4. Isolation of Extracellular Vesicles

Exosomes were extracted from the supernatant of the retinal organoids culture medium using a miRCURY exosome isolation kit (Qiagen, Hilden, Germany) according to the manufacturer’s instructions. In essence, an initial spin was performed at 3000× *g* for 10 min to remove cells and debris, then the medium was filtered. The corresponding amounts of reagents were added proportional to the organoid medium volume. The mixtures were vortexed and incubated at 4 °C overnight and then centrifuged at 10,000× *g* for 30 min at 20 °C, followed by exosome pellet resuspension in the manufacturer-supplied suspension buffer. The exosome pellets were resuspended in a 50 uL suspension buffer each with 1 mL starting volumes. All exosomes were stored at −80 °C immediately after isolation until further experimentation.

### 4.5. Characterization of Extracellular Vesicles

Nanoparticle tracking analysis (NTA) was performed to determine the concentration and size of organoid exosomes [69]. The exosomes were characterized using a NanoSight NS300 instrument equipped with NTA 3.4 analytical software and a 488-nm laser (Malvern Panalytical Ltd., Malvern, UK); at least five 30-s videos were recorded per sample in light scatter mode at a camera level of 11 to 13. The software settings for the analysis were kept constant for all measurements (screen gain = 10, detection threshold = 7). The exosomes were diluted in 0.22 µM filtered *phosphate-buffered* saline (PBS) to an appropriate concentration before analysis.

Exosomes derived from ROs were morphologically analyzed by transmission electron microscopy [70]. An exosome suspension diluted in PBS at a ratio of 1:10 was incubated on a 200-mesh sized, formvar/carbon-coated and charged nickel grid (Electron Microscopy Sciences, Hatfield, PA, USA) for 2 min. Next, the grid was fixed in 2.5% glutaraldehyde for 10 min and then washed thrice with 0.1 M PBS, followed by a brief contact of its surface with a water droplet. The grid was blotted to remove excess liquid and was then placed in 8 µL of 2% uranyl acetate for 2 min, subsequently examined, and imaged using a JEM-1400 flash electron microscope equipped with an AMT XR401 scientific complementary metal oxide semiconductor (sCMOS) camera and AMT Capture Engine software (JEOL Ltd., Tokyo, Japan).

Western blot analysis was used to determine the exosome protein markers [71]. The exosome proteins (5 ug/well) were loaded and electrophoresed on 10% polyacrylamide gels then transferred to nitrocellulose membranes at 80 V for 120 min. The membranes were blocked for 1 h at room temperature (RT) in 5% skim milk and washed in 0.1% Tween-20 in Tris-buffered saline (TBS), followed by incubation overnight at 4 °C with an anti-CD9 monoclonal antibody 1:500 dilution (Cell Signaling Technology, Danvers, MA, USA), anti-CD63 monoclonal antibody 1:1000 dilution (Santa Cruz Biotechnology, Dallas, TX, USA), anti-CD81 monoclonal antibody 1:500 dilution (Santa Cruz Biotechnology), anti-Calnexin monoclonal antibody 1:500 dilution (Cell Signaling Technology), and anti-β-actin monoclonal antibody 1:10,000 dilution (Sigma-Aldrich). The membranes were then incubated for 1 h at RT with a horseradish peroxidase-conjugated secondary antibody 1:5000 dilution (GenDEPOT, Baker, TX, USA). The target protein was detected by an enhanced chemiluminescence solution (Amersham Pharmacia Biotech, Buckinghamshire, UK) and visualized using the digital system Azure Biosystems C280 (Azure biosystems, Dublin, CA, USA).

### 4.6. Intravitreal Injections

Intravitreal injections were performed with 2 uL of Exo-ROs (1 × 10^10^ particles/mL) or 0.1% PBS into one eye of each RCS rat under anesthesia. The other eye was not treated and was used as a control. After inducing anesthesia and mydriasis, a scleral puncture site was carefully created posterior to the limbus with a 30-gauge beveled needle. The injection was conducted carefully through the puncture site using a NanoFil syringe with a 34-gauge blunt needle (World Precision Instruments Inc., Sarasota, FL, USA). The needle was kept in place for 15 s to allow full delivery.

### 4.7. In Vivo Study Design and Treatment Schedules

An in vivo study with RCS rats was designed with two different injection schedules (Figure 1A). In group 1, Exo-ROs or 0.1% PBS was intravitreally administered into one eye of each RCS rat at post-natal day (PND) 21. These animals were sacrificed 2 weeks later immediately after going through an electroretinogram (ERG) and optomotor response (OMR) test. In group 2, two serial intravitreal injections with the same materials were administered to one eye of each RCS rat at PND 21 and 35. Two weeks after the completion of injection schedules, an OMR test and ERG were conducted, and the animals were sacrificed for further analysis.

### 4.8. Electroretinography

A scotopic ERG was used to assess functional alterations in the rat retinal photoreceptors. The animals were kept in the dark room for 16 h before the ERG recording. All the recording procedures were conducted under dim red light (λ > 600 nm). Hydroxypropyl methylcellulose gel was applied to the corneas and covered with gold-ring contact electrodes, and reference and ground electrodes were placed subcutaneously in the tail and ear, respectively. To decrease the signal-to-noise ratio, responses were obtained by averaging 3 to 5 signals with a 15-s inter-stimulus interval. The signals were amplified and filtered through a digital band-pass filter ranging from 3 to 500 Hz to yield a- and b-waves. White-flash stimuli were delivered using a Ganzfeld stimulator (UTAS-3000; LKC Technologies, Gaithersburg, MD, USA). For the scotopic ERG recordings, dark-adapted rat eyes were exposed to flash intensities ranging from −2 to 1.5 cd·s/m^2^. The scotopic a-wave amplitude was measured from baseline to the negative peak, whereas that of the b-wave was measured from the trough of the a-wave to the following response peak. All animals (*n* = 6/group) were recorded before the commencement of the experimental procedures, for a baseline recording. On the day after the experiment, all animals were again recorded regardless of using the 660 light treatment paradigms.

### 4.9. Optomotor Response Test

OMR was measured with the OptoMotry system (Cerebral-Mechanics, Lethbridge, AB, Canada). RCS rats were adapted to ambient light for 30 min and then placed on the stimulus platform, surrounded by four computer monitors displaying black and white vertical stripe patterns. An event in which an RCS rat stopped moving and began tracking the stripe movements with reflexive head turns was counted as a successful visual detection. The detection thresholds were then obtained from OptoMotry software.

### 4.10. Primary Rat RPE Cell Culture

Primary cultures of rat RPE cells were established with Sprague Dawley rats. In summary, the cornea, lens, and retina were then removed, and the posterior eyecups were immersed in fresh Hank’s balanced salt solution (Thermo Fisher Scientific). The posterior eyecups were then incubated with Dispase (StemCell Technologies) at 37 °C for 30 min. The RPE cells were then released from the basement (Bruch’s) membrane by gentle aspiration and were harvested. The harvested RPE cells were plated on 35-mm Matrigel-coated dishes (Corning Life Sciences) in a ‘Miller’ medium (DMEM supplemented with 20% FBS, N1 medium supplement, 2 mM GlutaMAX, 250 mg/mL taurine, 10 ng/mL hydrocortisone, 13 ng/mL triiodothyronine, and 1% penicillin/streptomycin). The cells were subcultured when they reached confluence. For the H_2_O_2_ treatment groups, the cells were exposed to 200 uM H_2_O_2_ (Sigma-Aldrich) for 2 h. Following this, the cells were washed once with PBS, then 50 uL of green fluorescent dye-labeled Exo-ROs were added to the culture media and incubated for another 30 min.

### 4.11. Immunohistochemistry

RCS rats were sacrificed under deep anesthesia and perfused with 4% paraformaldehyde in 0.1 M PBS, and the eyeballs were enucleated. After removal of connective tissues, corneas, irises, and lens, the posterior eyecups were prepared for further morphological studies. The posterior eyecups were fixed in 4% paraformaldehyde for 1 h at 4 °C and dehydrated in 30% sucrose for 12 to 16 h at 4 °C. The posterior eyecups were embedded with optimal cutting frozen section compound (Leica, Wetzlar, Germany), and they were stored at −80 °C until further use. The eyecups were sectioned sagittally at 10 um in thickness and mounted onto a micro slide glass (Matsunami Glass Inc., Ltd., Osaka, Japan). All retinal sections were conducted following the dorsal–ventral axis and comprised both the superior and the inferior retina. The slides with the optic nerve head were dried for 30 min at 50 °C and washed with PBS and Tween-20 (PBST) for 15 min. The slides were blocked with 5% normal donkey serum (Jackson ImmunoResearch, West Grove, PA, USA) in PBST for 1 h and incubated with anti-rhodopsin 1:1000 (Millipore, Burlington, MA), anti-recoverin 1:1000 (Millipore), anti-M/L opsin 1:1000 (Millipore), anti-S-opsin 1:1000 (Millipore), p-p38 1:1000 (Cell Signaling Technology), phosphorylated c-Jun N-terminal kinase (p-JNK) 1:1000 (Cell Signaling Technology), and phosphorylated extracellular signal-regulated kinase 1/2 (p-ERK1/2) 1:1000 (Cell Signaling Technology) in 5% normal donkey serum overnight at 4 °C. Subsequently, they were washed with PBST and then incubated with secondary Alexa Fluor 568 donkey anti-rabbit IgG 1:1000 (Invitrogen) and Alexa Fluor 488 donkey anti-mouse IgG 1:1000 (Invitrogen) in 5% normal donkey serum for 2 h at RT. They were then washed and incubated with Hoechst 33,342 in PBS (Invitrogen) for 1 min at RT. The tissue sections were then washed and covered with coverslips using a mounting medium (Dako, Santa Clara, CA, USA) and imaged with a confocal microscope (Leica DMi8; Leica).

### 4.12. Terminal Deoxynucleotidyl Transferase dUTP Nick end Labeling (TUNEL) Assays

Apoptotic cell death in RCS rat retinas was determined using an In Situ Cell Death Detection Kit (Sigma-Aldrich) according to the manufacturer’s protocol. In essence, the cryosections of the eyecup were incubated with permeabilization solution (0.1% Triton X-100 and 0.1% sodium citrate), washed twice with PBS, and then incubated with a mixture of terminal deoxynucleotidyl transferase and fluorescein-labeled deoxyuridine triphosphate in a humidified atmosphere for 60 min at 37 °C in the dark. After washing with PBS, Hoechst 33,342 (H1399; Invitrogen) was used for nuclear staining. Fluorescence images were acquired using a laser confocal microscope (Leica DMi8; Leica).

### 4.13. Quantitative Reverse Transcription-Polymerase Chain Reaction (PCR)

The total RNA from RCS rat retinas was isolated using TRIzol reagent (Invitrogen), and the first-strand cDNA was synthesized using a SuperScript™ III First-Strand Synthesis System (Invitrogen) according to the manufacturer’s protocol. The cDNA was amplified using a QuantiSpeed SYBR Hi-ROX Kit (PhileKorea, Daejeon, Korea) on a StepOnePlus Real-Time PCR System (Applied Biosystems, Carlsbad, CA, USA). The thermocycling conditions were as follows: polymerase activation at 95 °C for 2 min, followed by 40 cycles of 95 °C for 5 s (denaturation), and 60 °C for 30 s (annealing/extension). The melt analysis was performed at 95 °C for 15 s, 60 °C for 1 min, and 95 °C for 15 s for each step. Glyceraldehyde 3-phosphate dehydrogenase was used as an internal control. The mRNA expression levels were quantified using the 2^−ΔΔCT^ method. The primer sequences used are listed in Table 1.

MAPK2K4, mitogen-activated protein kinase kinase 4; MAPK, mitogen-activated protein kinase; GAPDH, glyceraldehyde 3-phosphate dehydrogenase.

### 4.14. Western Blot Analysis

Retinal tissues and primary rat RPE cells were extracted with a whole cell extract buffer containing 10 mM HEPES (pH 7.9), 0.1 mM EDTA, 0.1 mM EGTA, 5% glycerol, 1 mM DTT, and 400 mM NaCl. Protein concentration was determined using a bicinchoninic acid protein assay kit following the manufacturer’s instruction (Thermo Fisher Scientific). Protein samples were mixed with an equal volume of 5 × sodium dodecyl sulfate (SDS) sample buffer, boiled for 5 min, and separated by 9% SDS-polyacrylamide gel electrophoresis (PAGE) gels. After electrophoresis, the proteins were transferred to polyvinylidene fluoride membranes. The membranes were blocked in 3% bovine serum albumin for 1 h. To evaluate the MAPK pathway cascade, specific antibodies against pERK 1:2000 dilution (Cell Signaling Technology), ERK 1:2000 dilution (Cell Signaling Technology), pJNK 1:1000 dilution (Cell Signaling Technology), JNK 1:2000 dilution *(Santa Cruz Biotechnology*, Santa Cruz, CA, USA), pp38 1:1000 dilution (Cell Signaling Technology), p38 1:2000 dilution (Cell Signaling Technology), cleaved caspase 3 1:1000 dilution (Cell Signaling Technology), and β-actin monoclonal antibody 1:10,000 dilution (Sigma-Aldrich) were used. The primary antibody was removed by washing the membranes three times in PBST and incubated for 1 h with horseradish peroxidase-conjugated anti-mouse or rabbit IgG 1:5000 dilution (GenDEPOT). Following this step of washing three times in PBST, immunopositive bands were visualized using a chemiluminescence reagent (ATTO Corp., Tokyo, Japan) and an azure imaging system (Azure biosystems). The band intensity was analyzed using ImageJ 1.53 software (National Institutes of Health, Bethesda, MD, USA).

### 4.15. RNA Sequencing and Gene Ontology-Pathway Enrichment Analysis

The total and exosomal RNAs were extracted using Trizol reagent (Invitrogen) and TRIzol LS reagent (*Ambion,* Carlsbad, CA, USA) respectively, according to the manufacturer’s instructions. RNA quality was assessed by Agilent 2100 bioanalyzer using the RNA 6000 Pico Chip (Agilent Technologies, Amstelveen, The Netherlands), and RNA quantification was performed using a NanoDrop 2000 Spectrophotometer system (Thermo Fisher Scientific).

The total RNA sequencing library was prepared using the NEBNext Ultra II Directional RNA-Seq Kit (New England BioLabs, Ipswich, MA, USA). The isolation of mRNA was performed using the Poly (A) RNA Selection Kit (Lexogen Inc., Vienna, Austria). The isolated mRNAs were used for the cDNA synthesis and shearing, following the manufacturer’s instructions. Indexing was performed using the Illumina indexes 1 to 12. The enrichment step was carried out using PCR. Subsequently, the libraries were checked using the TapeStation HS D1000 Screen Tape (Agilent Technologies) to evaluate the mean fragment size. Quantification was performed using the library quantification kit, a StepOne Real-Time PCR System (Life Technologies, Carlsbad, CA, USA). High-throughput sequencing was performed as paired-end 100 sequencing using NovaSeq 6000 (Illumina, Inc., San Diego, CA, USA). Quality control of raw sequencing data was performed using FastQC (https://www.bioinformatics.babraham.ac.uk/projects/fastqc/, 7 December 2021). The adapter and low-quality reads (<Q20) were removed using FASTX_Trimmer (http://hannonlab.cshl.edu/fastx_toolkit/, 7 December 2021) and BBMap (https://sourceforge.net/projects/bbmap/, 10 December 2021), respectively. Then, the trimmed reads were mapped to the reference genome using HISAT2 (http://daehwankimlab.github.io/hisat2/main/, 13 December 2021), and the read count was extracted using HTseq [72]. The data were processed based on the FPKM+Geometric normalization method, differential expression gene (DEG) analysis, and exactTest function using EdgeR within R (R development Core Team, 2020). The construction of a miRNA library was performed using the NEBNext Multiplex Small RNA Library Prep kit (New England BioLabs) according to the manufacturer’s instructions. Briefly, for the library construction, total RNA from each sample was used to ligate the adaptors, and then cDNA was synthesized using reverse transcriptase with adaptor-specific primers. PCR was performed for library amplification, and the libraries were cleaned up using QIAquick PCR Purification Kit (Qiagen) and the PAGE gel. The yield and size distribution of the small RNA libraries were assessed by the Agilent 2100 Bioanalyzer instrument for the High-sensitivity DNA Assay (Agilent Technologies). High-throughput sequences were produced by NextSeq500 system as a way of single-end 75 sequencing (Illumina, Inc.). The data analysis sequence reads were mapped by the bowtie2 software tool to obtain a bam file. The mature miRNA sequence was used as a reference for mapping, and the read counts mapped on the mature miRNA sequences were extracted from the alignment file using bedtools version 2.25.0 (Quinlan and Hall, 2010) [73] and Bioconductor [74], which use R statistical programming language (R development Core Team, 2016). The read counts were used to determine the expression level of miRNAs. However, the CPM+TMM normalization method was used for comparison between samples. The target genes of miRNAs were predicted using miRTarBase version 8.0 and TarBase v8.0 in miRNet (https://www.mirnet.ca, 28 Feburary 2022). Gene ontology and KEGG pathway analysis were performed using ClueGo packages [75] of Cytoscape software (version 3.9; https://cytoscape.org, 2 March 2022).

### 4.16. Statistical Analysis

All experiments were independently repeated at least thrice. The statistical significance of the differences was analyzed by a one-way analysis of variance (ANOVA), two-way ANOVA, or paired *t*-test using GraphPad Prism 9 software (GraphPad Software, Inc., San Diego, CA, USA). The significant differences are indicated by a single (*^,#^; *p* < 0.05), double (**^, ##^; *p* < 0.01), or triple asterisk (***^,###^; *p* < 0.001).

### 4.17. eTOC Summary

Intravitreal administration of exosomes extracted from ROs can reduce photoreceptor apoptosis, prevent outer nuclear layer (ONL) thinning, and preserve visual function in Royal College of Surgeons rats, while suppressing the expression of MAPK signaling pathway related genes and proteins.

## Figures and Tables

**Figure 1 ijms-24-12068-f001:**
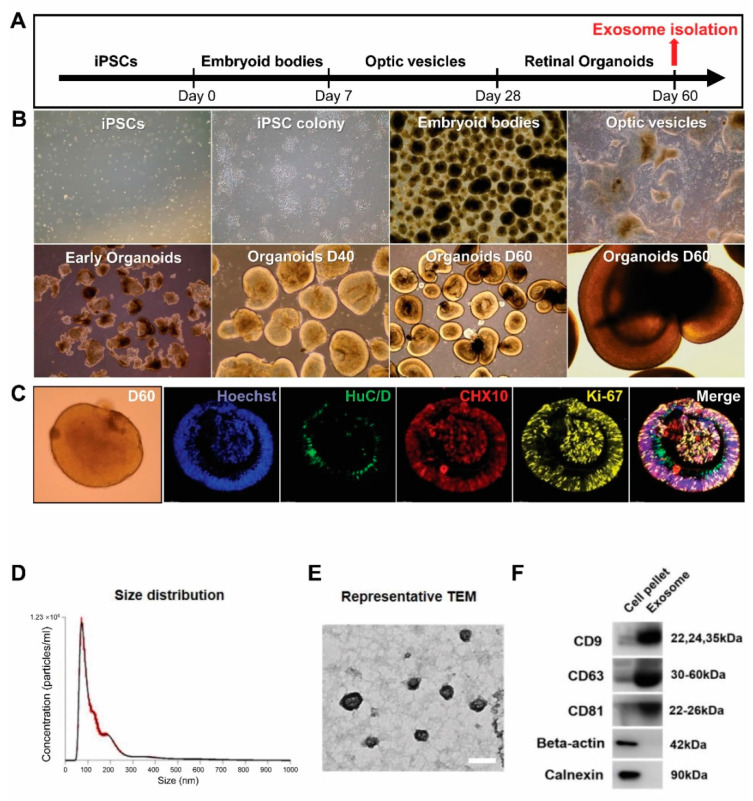
Generation of human induced pluripotent stem cell (hiPSC)-derived retinal organoids (ROs) and characterization of exosomes from ROs. (**A**) Schematic overview of the differentiation protocol. Conditioned medium is collected at 60 days (D60) of RO differentiation. (**B**) Main steps of hiPSC-derived RO development in vitro: hiPSC colony, embryoid body formation, optic vesicles, and ROs. (**C**) Immunofluorescence staining of human neuronal protein (green) in retinal ganglion cells and CHX10 (red) and Ki-67 (yellow) in retinal progenitors at D60 of differentiation (scale bar = 20 µm). (**D**) Diameter and concentration distribution of exosomes from hiPSC-derived RO conditioned media at D60. (**E**) Transmission electron microscopy images of exosomes from hiPSC-derived ROs (scale bar = 100 µm). (**F**) Western blot analysis of the expression of the exosomal markers CD63, CD81, and CD9 in exosomes from hiPSC-derived ROs.

**Figure 2 ijms-24-12068-f002:**
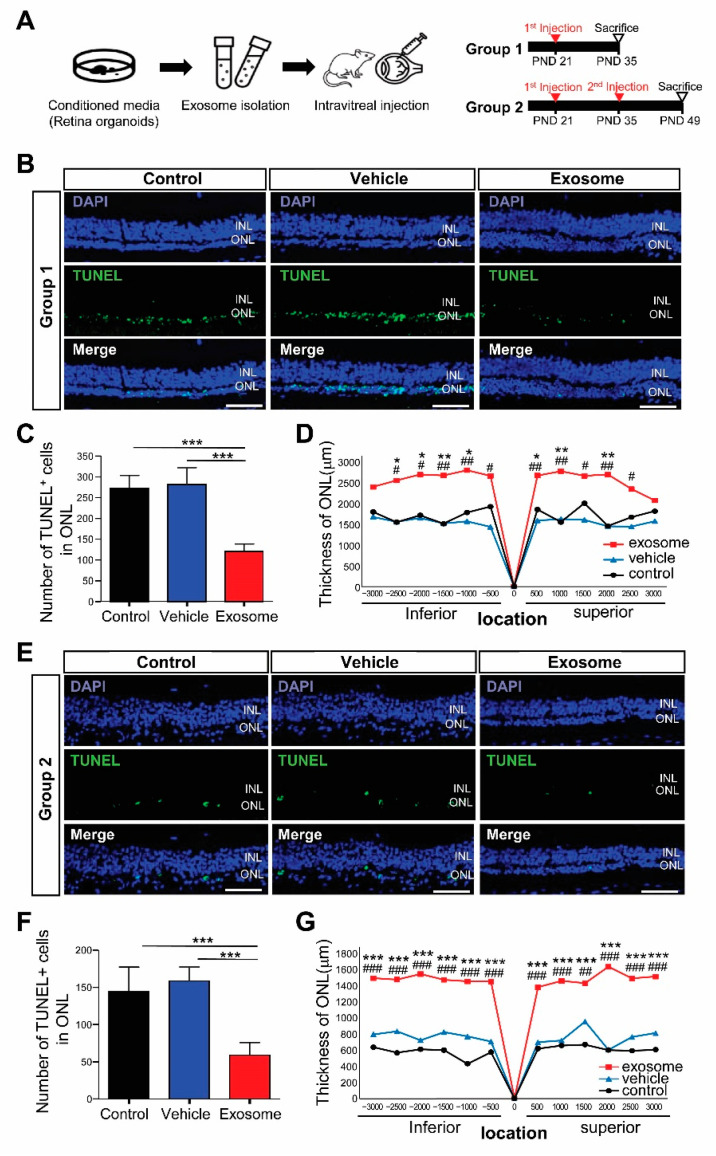
Exosomes from retinal organoids (Exo-ROs) alleviate photoreceptor cell death and preserve visual function in the Royal College of Surgeons (RCS) rats. (**A**) Schematic diagram of the in vivo RCS rat experiment design. Group 1, single-injection of Exo-ROs on postnatal day (PND) 21; group 2, twice-injection on PNDs 21 and 35. (**B**) Terminal deoxynucleotidyl transferase dUTP nick end labeling (TUNEL) staining in control, vehicle-, and Exo-RO-treated retinas in group 1 (scale bar = 20 µm). (**C**) Comparison of the numbers of TUNEL-positive cells in the whole retina of control, vehicle-, and Exo-RO-treated rats in group 1 (*n* = 5 rats per group; * *p* < 0.05, ** *p* < 0.01, *** *p* < 0.001, by one-way analysis of variance (ANOVA)). (**D**) Comparison of outer nuclear layer (ONL) thickness in control, vehicle-, and Exo-RO-treated retinas at different locations and distances from the optic nerve head (ONH) in group 1 (*n* = 3 per group, * *p* < 0.05, ** *p* < 0.01, *** *p* < 0.001 between control and exosome; ^#^
*p* < 0.05, ^##^
*p* < 0.01, ^###^
*p* < 0.001, between vehicle and exosome, by two-way ANOVA). (**E**) TUNEL staining in control, vehicle-, and Exo-RO-treated retinas in group 2 (scale bar = 20 µm). (**F**) Comparison of the numbers of TUNEL-positive cells in the whole retinas of control, vehicle-, and Exo-RO-treated rats in group 2 (*n* = 5 rats per group; * *p* < 0.05, ** *p* < 0.01, *** *p* < 0.001, by one-way ANOVA). (**G**) Comparison of ONL thickness in control, vehicle-, and Exo-RO-treated retinas at different locations and distances from the ONH in group 2 (*n* = 3 per group; * *p* < 0.05, ** *p* < 0.01, *** *p* < 0.001 between control and exosome; ^#^
*p* < 0.05, ^##^
*p* < 0.01, ^###^
*p* < 0.001, between vehicle and exosome, by two-way ANOVA).

**Figure 3 ijms-24-12068-f003:**
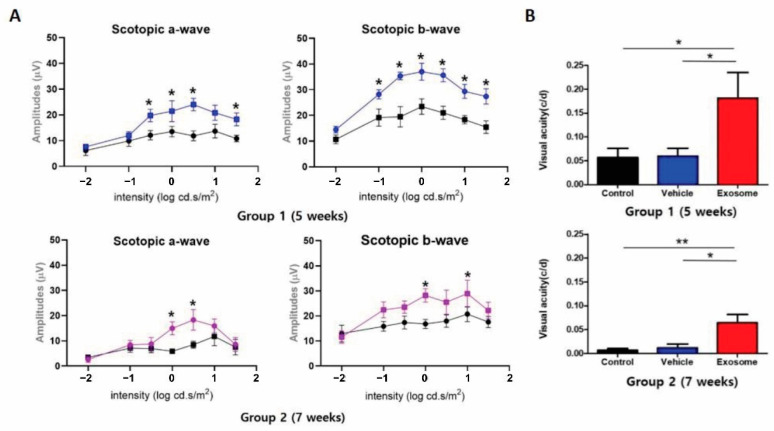
(**A**) Quantitative analyses of amplitude changes of a- and b-waves by different light intensity stimulus (*n* = 6 eyes per group) (* *p* < 0.05, by paired *t*-test). (**B**) Visual acuity measured through the optomotor response test in control, vehicle-, and Exo-RO-treated retinas of group 1 and 2. (*n* = 6 per group; * *p* < 0.05, ** *p* < 0.01, by two-way ANOVA).

**Figure 4 ijms-24-12068-f004:**
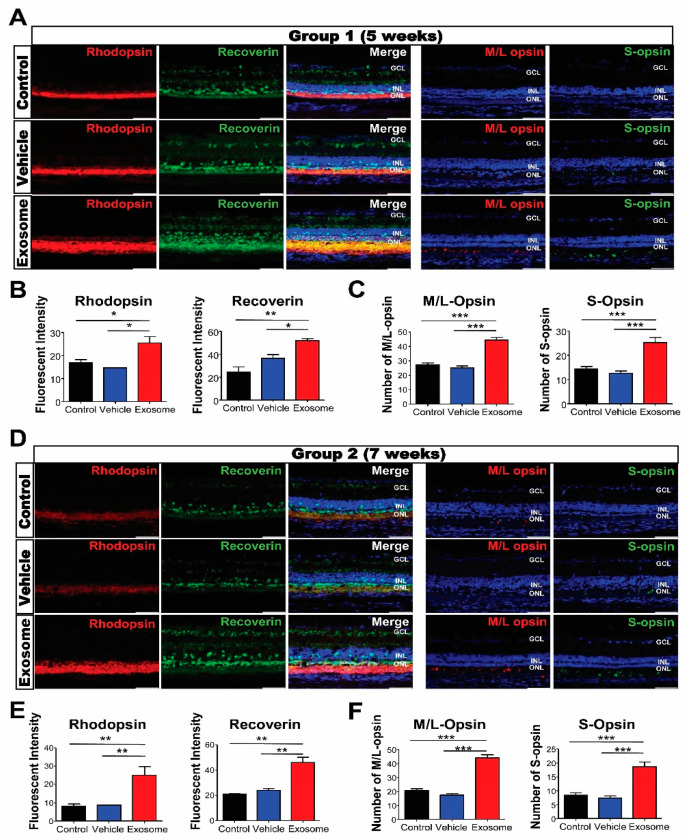
Exosomes from retinal organoids (Exo-ROs) protect photoreceptor cells (cone and rod) in the Royal College of Surgeons rats. (**A**) Representative images of immunofluorescence for specific photoreceptor markers, including recoverin (green), rhodopsin (red), middle/long (M/L) opsin (red), and sensitive (S) opsin (green), in group 1 (scale bar = 50 µm). (**B**) Quantification of recoverin and rhodopsin fluorescence. The fluorescence intensities of rhodopsin and recoverin are significantly higher in Exo-RO-treated retinas compared with those in the control or vehicle-treated retinas of group 1. (**C**) Quantification of M/L and S opsins. The numbers of M/L and S opsins are significantly elevated in the Exo-RO-treated retinas compared with those in the control or vehicle-treated retinas of group 1. (**D**) Representative images of immunofluorescence for specific photoreceptor markers, including recoverin (green), rhodopsin (red), M/L opsin (red), and S opsin (green), in group 2 (scale bar = 50 µm). (**E**) Quantification of recoverin and rhodopsin fluorescence. The fluorescence intensities of rhodopsin and recoverin are significantly higher in Exo-RO-treated retinas than those in the control or vehicle-treated retinas of group 2. (**F**) Quantification of M/L and S opsins. The numbers of M/L and S opsins are significantly elevated in the Exo-RO-treated retinas than those in the control or vehicle-treated retinas of group 2 (* *p* < 0.05, ** *p* < 0.01, *** *p* < 0.001, by one-way analysis of variance).

**Figure 5 ijms-24-12068-f005:**
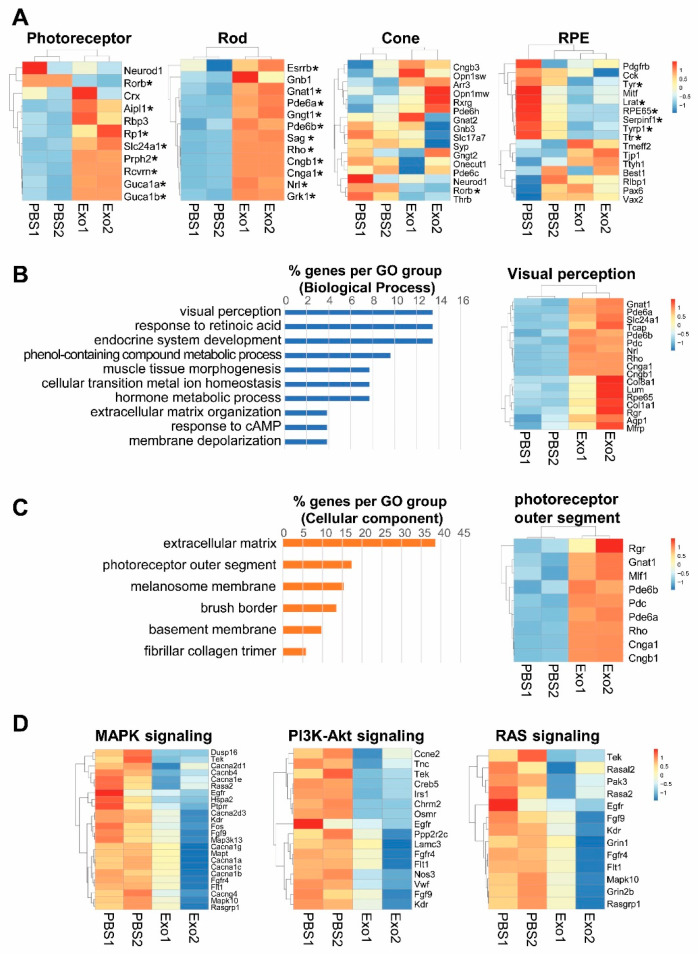
Changes in the gene expression profile of the Royal College of Surgeons (RCS) rats after exosomes from retinal organoids (Exo-ROs) treatment. Total RNA is analyzed in the eyes of RCS rats before and after Exo-RO treatment using RNA-seq. (**A**) Transcriptional profiling of retinal cell markers. Exo-RO treatment significantly upregulates the expression of photoreceptors, rod cells, cone cells, and retinal pigment epithelium cells (* FDR *q* < 0.05). (**B**,**C**) Gene ontology analysis of upregulated genes in Exo-RO-treated retinas. (**B**) The genes are mainly enriched in ‘endocrine system development’, ‘response to retinoic acid’, and ‘visual perception’ for biological processes. (**C**) The genes are mainly enriched in ‘extracellular matrix’, ‘photoreceptor outer segment’, and ‘melanosome membrane’ for cellular components. (**D**) The expression of inflammation-related pathways, including mitogen-activated protein kinase (MAPK), phosphoinositide 3-kinase-Akt, and RAS signaling, is significantly downregulated in Exo-RO-treated retinas (logFC < −1, false discovery rate [*q*-value] < 0.05).

**Figure 6 ijms-24-12068-f006:**
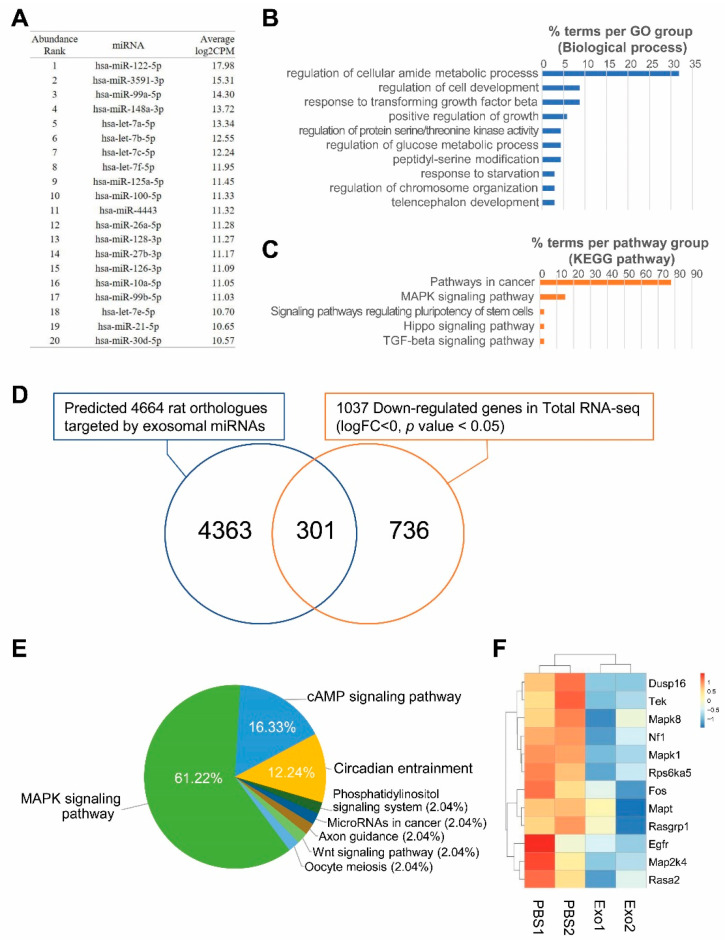
Exosomes from retinal organoids (Exo-ROs) contain microRNAs (miRNAs) mainly involved in the mitogen-activated protein kinase (MAPK) signaling pathway. (**A**) The 20 most abundant miRNAs in Exo-ROs with more than 1000 counts per million (cpm). (**B**) Gene ontology and (**C**) pathway enrichment of the 293 genes are predicted as shared targets of 10 or more miRNAs among the 20 most abundant miRNAs. (**D**) Extraction of actual target genes by comparison of predicted target genes and differentially expressed genes in Exo-RO-treated the Royal College of Surgeons (RCS) rats. Of the 4929 human target genes, 4674 rat orthologs are identified. Among them, 301 genes are significantly downregulated in Exo-RO-treated eyes relative to those in vesicle-treated eyes (*p* < 0.05, logFC < 0). (**E**) The Kyoto Encyclopedia of Genes and Genomes pathway analysis shows that the 301 genes are mainly involved in the MAPK signaling pathway (61.22%). (**F**) Significantly downregulated genes of MAPK pathway, which are targeted by top-20 abundant miRNAs in Exo-ROs (cpm > 1000), in retina of Exo-ROs-treated RCS rat (false discovery rate (*q*-value) < 0.05).

**Figure 7 ijms-24-12068-f007:**
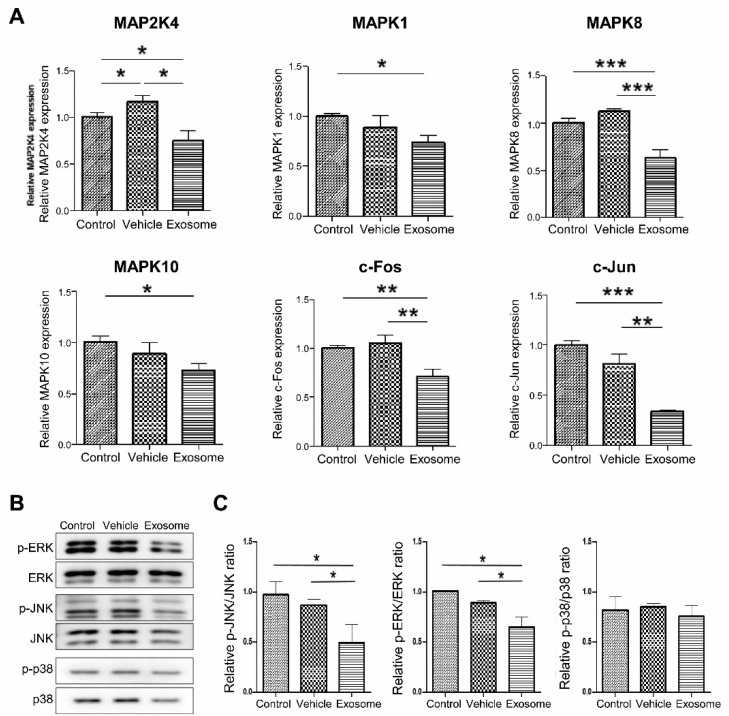
Assessment of mitogen-activated protein kinase (MAPK) signaling inhibition by exosomes from retinal organoids (Exo-ROs) in vivo. (**A**) Reverse transcription-quantitative polymerase chain reaction analysis identifies the key gene expressions of the MAPK pathway, including those of *MAP24*, *MAPK1*, *MAPK8*, *MAPK10*, *c-FOS*, and *c-JUN*. These genes are significantly lower in the Exo-RO-treated retinas than those in control and vesicle-treated retinas. (**B**) Quantitative analysis of band density. Each experiment is executed in triplicate. Data are shown as mean ± standard error (*n* = 3). (**C**) The expression of phosphorylated MAPK family members is assessed by Western blotting after treatment with Exo-ROs. The expression of phosphorylated c-Jun N-terminal kinase (p-JNK)/JNK and phosphorylated extracellular signal-regulated kinase (p-ERK)/ERK is significantly decreased in Exo-RO-treated retinas compared to control and vehicle-treated retinas (* *p* < 0.05, ** *p* < 0.01, *** *p* < 0.001, by one-way analysis of variance).

**Figure 8 ijms-24-12068-f008:**
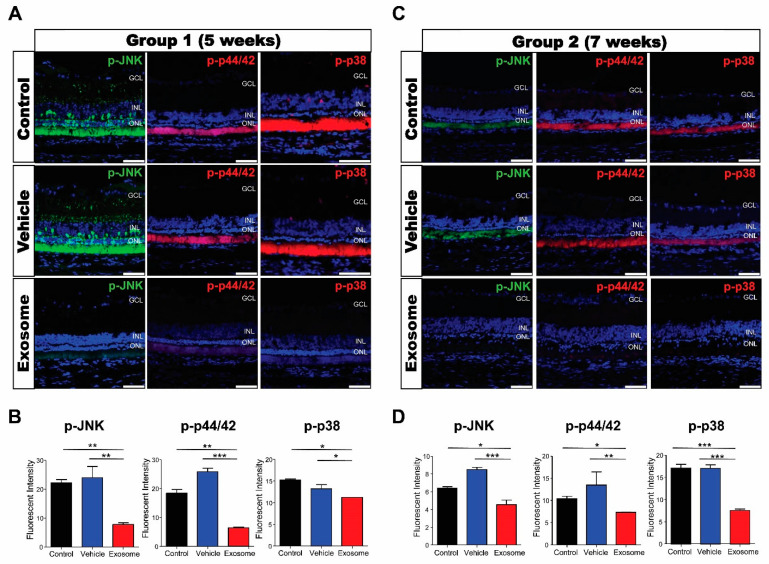
Exosomes from retinal organoids (Exo-ROs) markedly inhibit mitogen-activated protein kinase (MAPK) signaling in the Royal College of Surgeons rats. (**A**) Representative immunofluorescence images for specific MAPK pathway markers in group 1 (scale bar = 50 µm). (**B**) Quantification of p-JNK, p-44/42, and p-p38 fluorescence intensities. The fluorescence intensities of phosphorylated c-Jun N-terminal kinase (p-JNK), p-44/42, and p-p38 are significantly lower in Exo-RO-treated retinas than those in control or vehicle-treated retinas of group 1. (**C**) Representative immunofluorescence images for specific MAPK pathway markers in group 2 (scale bar = 50 µm). (**D**) Quantification of p-JNK, p-44/42, and p-p38 fluorescence intensities. The fluorescence intensities of p-JNK, p-44/42, and p-p38 are significantly lower in Exo-RO-treated retinas than those in control or vesicle-treated retinas of group 2 (* *p* < 0.05, ** *p* < 0.01, *** *p* < 0.001, by one-way analysis of variance).

**Figure 9 ijms-24-12068-f009:**
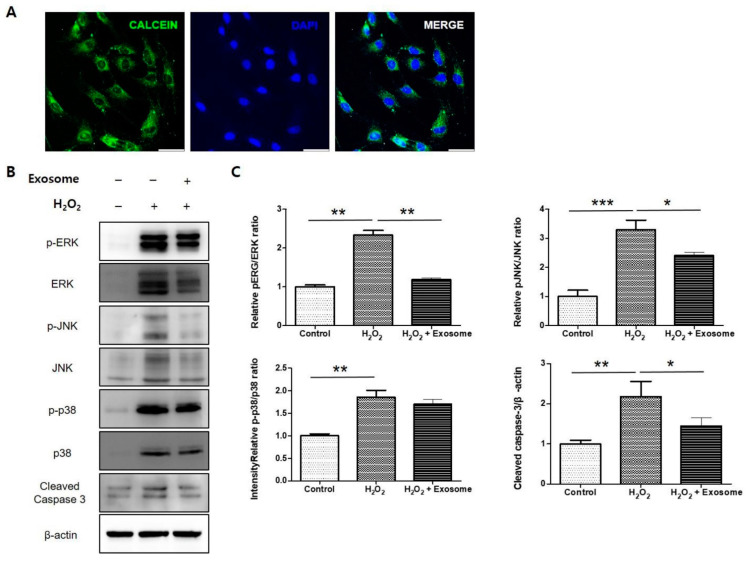
Effect of exosomes from retinal organoids (Exo-ROs) on primary rat retinal pigment epithelium (RPE) cells following H_2_O_2_ mediated oxidative stress injury. (**A**) Uptake of Exo-ROs by RPE cells. Green fluorescein particles are observed throughout the cell cytoplasm. (**B**) Quantitative analysis of band density. Each experiment is executed in triplicate. Data are shown as mean ± standard error (*n* = 3) (**C**) Western blot analysis of phosphorylated mitogen-activated protein kinase (MAPK) and cleaved caspase-3 in H_2_O_2_-treated RPE cells treated with Exo-ROs. Exo-ROs treatment suppresses the expression of phosphorylated c-Jun N-terminal kinase (p-JNK)/JNK, p-44/42 and cleaved caspase-3 compared with that in the H_2_O_2_-treated RPE cells (* *p* < 0.05, ** *p* < 0.01, *** *p* < 0.001, by one-way analysis of variance). RCS, Royal College of Surgeons; ONL, outer nuclear layer.

**Table 1 ijms-24-12068-t001:** List of primers used for quantitative reverse transcription-polymerase chain reaction.

Gene	Forward 5′-3′	Reverse 5′-3′
MAPK2K4	AGAGACTGAGAACCCACAGCAT	CTACTCCGCATCACTACATCCA
MAPK1	TGTTGCAGATCCAGACCATG	CAGCCCACAGACCAAATATCA
MAPK8	ATTTGGAGGAGCGAACTAAG	CTGCTGTCTGTATCCGAGGC
MAPK10	TCGAGACCGTTTCAGTCCAT	CCACGGACCAAATATCCACT
c-FOS	GTCCGTCTCTAGTGCCAACTTTAT	GTCTTCACCACTCCCGCTCT
c-Jun	TCCCCCATCGACATGGAGTCTC	CCAGTCCATCTTGTGTACCCTTG
GAPDH	CTGAGTATGTCGTGGAGTCTA	CTGCTTCACCACCTTCTTGAT

MAPK2K4, mitogen-activated protein kinase kinase 4; MAPK, mitogen-activated protein kinase; GAPDH, glyceraldehyde 3-phosphate dehydrogenase.

## Data Availability

Data will be made available on request.

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
