# Peer review of "Intravitreal Administration of Retinal Organoids-Derived Exosomes Alleviates Photoreceptor Degeneration in Royal College of Surgeons Rats by Targeting the Mitogen-Activated Protein Kinase Pathway"

_ijms, 2023, doi:10.3390/ijms241512068_

Round 1

Reviewer 1 Report

This paper investigated the effect of intravitreal injection of  exosomes produced by 60-d old pluripotent stem cell-derived retinal organoids on retinal degeneration in the RCS rat. . The effect of the exosomes in vivo was tested in 2 scenarios: (1) injection on P21 and sacrifice 2 weeks later (P35) ; (2) injection on P21 and P35, sacrifice 2 weeks later (on P49). Visual function was tested by electroretinograms (ERGs) and optokinetic testing prior to sacrifice. There was a clear rescue  of photoreceptors over a wide area. The effect of exosomes was also tested in primary RPE cell cultures treated with H2O2.

In addition, the authors performed molecular analysis (qPCR and RNAseq) to determine the mechanism of action. The data demonstrate the involvement of the MAPK pathway.

General comments:

Overall the manuscript is well written; and the data look convincing. The reviewer’s only concern is with some of the figures which have too many data crammed into too little space (see detailed comments below)..

Detailed comments:

Figures 2-5 and 7: contain too much detail so that the panels appear too small.  The resolution of the original tiff images is acceptable, but the figure panels may appear too small in the published journal. Is there any way to move some of this to supplements, or to split up some figures so the panels can be larger?

Author Response

Figures 2-5 and 7: contain too much detail so that the panels appear too small.  The resolution of the original tiff images is acceptable, but the figure panels may appear too small in the published journal. Is there any way to move some of this to supplements, or to split up some figures so the panels can be larger?

->We thank the reviewer for the feedback and appreciate the suggestion to improve our manuscript. Following the reviewer’s suggestion, we have split some of the figures and rearranged some as Supplements. In addition, figure 4 and 8 contain numerous figure panels, so we removed some of figure panels including DAPI counterstaining images and enlarge the figure panels to show them more clearly. We expect that this adjustment will enhance the visibility of the figure panels for readers.

Reviewer 2 Report

The manuscript submitted by Han et al., was designed, proved and very well written. The findings of the study useful for development of cell based and gene-based therapies for the eye disorders.  

The introduction of the first paragraph – statements were repetitive. Add relevant references for the proposed statements.

Write a note on RD statistics and role of cell-based therapies in the ocular disease over conventional treatment approaches.

Why the Intravitreal route selected for the testing of the EX-ROs not discussed in the manuscript.

Materials section missing in the manuscript.

Write the ethical committee approval number in the animal section. Write the age of the animal used in the study.

Line 127 – The ROs were differentiated as described previously [31]. Rewrite the sentence.

Write the supportive references for section 2.6 methods.

Figure 1e – the scale bar of the TEM image was very high. EVs reported with 100 nm particle size, but TEM studies showed 100 um. Recheck and justify?

The quality of the figures needs to be improved for better clarity.

Only minor edits needed.

Author Response

Reviewer 2

The manuscript submitted by Han et al., was designed, proved and very well written. The findings of the study useful for development of cell based and gene-based therapies for the eye disorders. 

  1. The introduction of the first paragraph – statements were repetitive. Add relevant references for the proposed statements.

-> First of all, thank you for the time you have put reviewing our manuscript and for your incisive and valuable comments. According to the reviewer’s suggestion, we have revised the repetitive statements and added relevant references to the first paragraph of Introduction section. (Lines 62-67)

Write a note on RD statistics and role of cell-based therapies in the ocular disease over conventional treatment approaches.

Thank you for the valuable comment. We have added the information as suggested into the discussion section.

Line 554–558: Recent statistical studies and advancements in cell-based therapies complement traditional approaches in the treatment of ocular diseases. These therapies hold immense potential for restoring vision and enhancing the quality of life for patients. The transplantation of stem cells is one of the potential therapeutic options for rescuing vision loss associated with RD.

Why the Intravitreal route selected for the testing of the EX-ROs not discussed in the manuscript.

-> In this study, we have demonstrated that the intravitreal injection of exosomes led to effective inhibition of photoreceptor cell death. Since the effect of exosomes does not last for a long period of time, repeated injections are likely to be required to sustain the therapeutic effect of exosomes when used for real patients. However, subretinal injection, regarded as one of the best strategies for gene therapy or stem cell transplantation, is technically far more demanding than intravitreal injection. For these reasons, subretinal injection is difficult to perform precisely and repeatedly on patients in clinical practice. Therefore, intravitreal injection was chosen for its technical feasibility. We have provided detailed information on these findings in the Discussion section. (Lines 564-568)

Materials section missing in the manuscript.

-> We apologize if we have misunderstood the question, but we believe it pertains to the adequacy of the explanation of the materials used in this study in the Methods section. The strain of rat, the origin of cells, reagents and anti-bodies used in present study are described in detail in Material and Methods part on page 6. If we did not fully understand this question, we deeply apologize for the misunderstanding, and kindly request further clarification to address your concerns more accurately.

Write the ethical committee approval number in the animal section. Write the age of the animal used in the study.

-> Thank you for the constructive comment. We have added the committee approval number in Animal section as suggested. (Line 121) For this study, we used 3-week-old RCS/Kyo rats, a stage at which photoreceptor death begins. We divided the rats into 2 groups: in group 1, exosomes were injected at postnatal day (PND) 21, and in group 2, they were injected twice at PND 21 and 35, respectively. After a 14-day injection period, we assessed the therapeutic effect of the exosomes in RCS rat. The animal study design is described in detail in Section 2.8 of the Methods.

Line 127 – The ROs were differentiated as described previously [31]. Rewrite the sentence.

->We thank the reviewer for the comment. The sentence was revised as suggested.

Lines 145-146 : The ROs were differentiated from ATCC DYR0100 hiPSCs according to a retinal differentiation protocol recently described by Lee et al.

Write the supportive references for section 2.6 methods.

-> We thank the reviewer for the comment. In accordance with the reviewer’s suggestion, we have added the references in 2.6 method section. (Lines 189, 197 and 206)

Figure 1e – the scale bar of the TEM image was very high. EVs reported with 100 nm particle size, but TEM studies showed 100 um. Recheck and justify?

-> We appreciate the reviewer’s comments. Following the reviewer’s suggestion, we replaced the TEM images with the scale bar of appropriate size to show the images clearly.

The quality of the figures needs to be improved for better clarity.:

 We appreciate the reviewer’s comment. As the reviewer have pointed out, we revised the figure with images of improved quality to show the effect of Exo-ROs more clearly.

Round 2

Reviewer 2 Report

No further comments.